# Human evolved regulatory elements modulate genes involved in cortical expansion and neurodevelopmental disease susceptibility

Hyejung Won [1,4], Jerry Huang[1], Carli K. Opland[1,4], Chris L. Hartl[1] & Daniel H. Geschwind [1,2,3]

Modern genetic studies indicate that human brain evolution is driven primarily by changes in gene regulation, which requires understanding the biological function of largely non-coding gene regulatory elements, many of which act in tissue specific manner. We leverage chromatin interaction profiles in human fetal and adult cortex to assign three classes of human-evolved elements to putative target genes. We find that human-evolved elements involving DNA sequence changes and those involving epigenetic changes are associated with human-specific gene regulation via effects on different classes of genes representing distinct biological pathways. However, both types of human-evolved elements converge on specific cell types and laminae involved in cerebral cortical expansion. Moreover, human evolved elements interact with neurodevelopmental disease risk genes, and genes with a high level of evolutionary constraint, highlighting a relationship between brain evolution and vulnerability to disorders affecting cognition and behavior. These results provide novel insights into gene regulatory mechanisms driving the evolution of human cognition and mechanisms of vulnerability to neuropsychiatric conditions.

[1] Neurogenetics Program, Department of Neurology, David Geffen School of Medicine, University of California, Los Angeles, Los Angeles, CA 90095, USA. [2] Semel Institute, David Geffen School of Medicine, University of California, Los Angeles, Los Angeles, CA 90095, USA. [3] Department of Human Genetics, David Geffen School of Medicine, University of California, Los Angeles, Los Angeles, CA 90095, USA. [4] Present address: Department of Genetics and UNC Neuroscience Center, University of North Carolina, Chapel Hill, NC 27599, USA. Correspondence and requests for materials should be addressed to D.H.G. (email: dhg@mednet.ucla.edu)

Human evolution is hypothesized to be driven primarily by changes in gene regulation rather than divergence in protein-coding sequences[1]. Recent comparative genomic and epigenomic studies have identified regions on the human lineage having either an accelerated sequence, referred to as human accelerated regions (HARs)[2–7], or epigenetic changes, referred to as human gained enhancers (HGEs)[8,9]. One class of human evolved genomic elements, HARs, are enriched in developmental enhancers, suggesting that they may drive evolution of human-specific traits via developmental gene regulation. Recent targeted sequencing of HARs in consanguineous autism spectrum disorder (ASD) families identified significant enrichment of rare bi-allelic variants, highlighting a potential role of HARs in susceptibility to neurodevelopmental disorders[10]. While some preliminary functional characterization has been conducted[10], accurately mapping the target genes regulated by these enhancers requires tissue specificity, since the majority of chromatin interactions are predicted to be highly tissue specific[11]. We sought to bridge the gap between genetic changes on the human lineage and the molecular basis of human evolution by mapping these genomic elements to their putative target genes using chromatin conformation in human brain[11].

We find that although HARs and HGEs target different genes and molecular pathways and exhibit different developmental trajectories, they converge in terms of their cell-type enrichment patterns. These patterns highlight that these elements regulate specific genes involved in primate cortical expansion enriched in neural progenitors of the outer subventricular zone (oSVZ), and their progeny, supragranular neurons, providing a regulatory map for understanding the molecular mechanisms underlying human cortical expansion. Both forms of regulatory elements also converge on genes that are highly conserved at the protein level, consistent with the model that noncoding regulatory elements drive evolutionary divergence by regulation of essential, highly constrained transcripts. Constrained genes targeted by these human evolved elements are enriched among those in which protein-disrupting mutations cause several neurodevelopmental conditions, including ASD. Finally, we use CRISPR activation in primary human neural progenitors to validate the functional impact of HARs predicted to regulate three highly conserved genes involved in brain patterning, *GLI2*, *GLI3*, and *TBR1*.

## Results

**HARs are enriched in regulatory elements of the fetal brain**. To more precisely identify the developmental window and tissue in which HARs play a regulatory role, we compared a previously compiled list of 2737 HARs[10] with DNase I hypersensitivity sites (DHS) in 51 cell/tissue types[12] (Methods). Consistent with previous results, HARs were significantly enriched in putative regulatory elements active prenatally (fetal adrenal gland, brain, kidney, lung, and muscle)[4], the strongest enrichment being observed in fetal brain (Fig. 1a, Supplementary Fig. 1a). We observed strong enrichment in predicted enhancer states, as well as H3K27ac, H3K4me1, and H3K4me2 marks, indicating that HARs are enriched in developing brain enhancers, more so than in adult brain (Supplementary Fig. 1b, c). Further, HARs were enriched in regulatory elements that were significantly more likely to be accessible in the germinal (neurogenic) zones of the developing cortex[13], highlighting their potential role in cortical neurogenesis (Supplementary Fig. 1d).

**HARs potentially regulate human brain development**. Although this analysis identified the tissue and stage where HARs are predicted to be the most active, it did not identify which genes are regulated by these elements. So, we next

integrated these data with a recently defined three-dimensional chromatin interaction map in developing human cortex, which identified physical enhancer-promotor/gene interactions[11]. We were able to assign 638 and 717 HARs to 972 and 1021 genes using contacts defined by three-dimensional chromatin conformation in cortical plate (CP, neocortical laminae containing post-mitotic neurons) and germinal zone (GZ, neocortical laminae consisting primarily of neural progenitors) in the fetal brain, respectively (Methods; Supplementary Fig. 2)[14]. When combined with intragenic HARs, we were able to assign 1028 HARs to 1648 putative target genes (Supplementary Tables 1–2, Supplementary Fig. 2, subsequently referred to as putative target genes for HARs). Only 26.3% of these physically interacting genes were the genes nearest to a HAR (Fig. 1b), which indicates how risky it is to perfunctorily assign regulatory elements to the nearest gene without other evidence. This is also consistent with emerging evidence that chromatin interactions often link genes to quite distal regulatory elements[11,15,16] and are not related to linear distance or genetic recombination, as defined by linkage disequilibrium[17].

The putative target genes of HARs are enriched for genes that regulate pathways involved in human brain development, regionalization, dorsal-ventral patterning, cortical lamination, and proliferation of neuronal progenitors (Fig. 1d, Supplementary Fig. 3), suggesting that multiple aspects of human brain development are subject to human-specific regulation. This includes genes driving the dorsal–ventral patterning of the telencephalon (*EMX2*, *PAX6*, *GLI3*, *NKX6.1*, and *NKX6.2*), genes playing major roles in cortical neurogenesis (*PAX6*, *HES1*, *SOX2*, *GLI3*, and *TBR2*), genes that specify laminar identity of cortical neurons (*TBR1*, *CUX1*, *POU3F2*, *POU3F3*, *RORB*, *MDGA1* and *ETV1*), and genes involved in axonal pathfinding (*DSCAML1* and *ROBO*). While the majority of genes are involved in forebrain development, a few putative target genes regulate mid- or hind-brain development, including *GLI2*, *EN1*, and *GBX2*.

Doan et al.[10] performed targeted sequencing of HARs in consanguineous families containing probands diagnosed with ASD, finding enrichment of rare sequence variants within HARs in patients with ASD. They also used chromatin interaction profiles in multiple non-neuronal cell types to identify putative target genes of HARs, a portion (20.8%) of which overlap with targets identified based on Hi-C from developing brain[11]. Although this overlap is significant ($P = 2.09 \times 10^{-17}$, OR = 2.5, Fisher's exact test) and implicates a high confidence core set of target genes (Supplementary Table 1), most target genes are discordant (Fig. 1c), which may be due to differences between tissue specific gene regulation, or methodologic factors.

To reduce confounding due to use of different analytic pipelines, we next examined genes that interact with HARs in non-neuronal cell types (embryonic stem cells: ES cells and fetal lung fibroblasts: IMR90 cells) using the same analytic pipeline previously applied to fetal brain[18,19]. We found that HARs interacted with 823 and 467 protein-coding genes in ES cells and IMR90 cells, respectively (Supplementary Fig. 4a, b). The majority of genes (~60%) interacted with HARs in a cell type-specific manner, again highlighting the cell type specific nature of chromatin interactions, consistent with the notion of tissue specific gene regulation[11,19] (Supplementary Fig. 4). Notably, many genes known to play major roles in cerebral cortex development and dorsal–ventral/anterior–posterior pattern specification, including *SOX2*, *PAX6*, *POU3F2*, *GLI3*, *EN1*, and *TBR2*, interacted with HARs in the developing cortex (Supplementary Fig. 4c), consistent with the model that chromatin contact maps in developing brain will likely provide more biologically relevant targets for human brain evolution than other tissues.

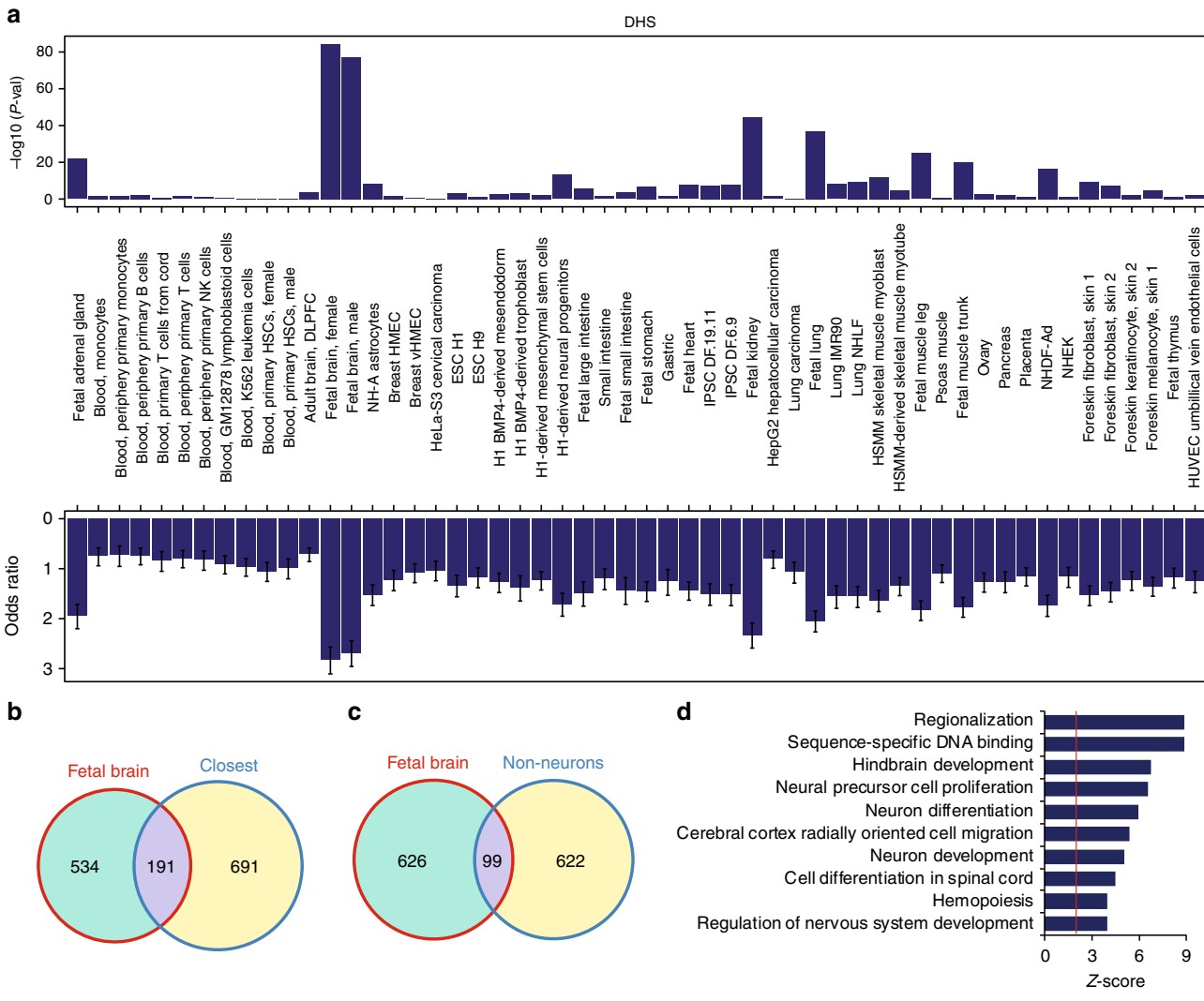

**Fig. 1** HARs interact with genes that regulate human brain development. **a** DHS enrichment of HARs in different cell/tissue types after controlling for evolutionary conservation (Methods). HSCs, hematopoietic stem cells; NK cells, natural killer cells; DLPFC, dorsolateral prefrontal cortex. *P*-values and odds ratio calculated by Fisher's exact test. Error bars denote for 95% confidence intervals (CI). **b** Overlap between nearest genes to HARs (closest) with genes that interact with HARs in developing brain (fetal brain). **c** Overlap between genes that interact with HARs in developing brain (fetal brain), and non-neuronal cell types (non-neurons)[10]. Protein-coding genes were used for Venn diagrams. **d** Gene ontology enrichment for HAR-associated genes

**Comparing different classes of human evolved elements**. We next analyzed another major type of regulatory element predicted to play a role in human brain evolution – the class of human gained enhancers (HGEs) and human lost enhancers (HLEs) – genomic regions that exhibit increased and decreased enhancer activity, respectively, as assessed through changes in active epigenetic marks (H3K27ac) on the human lineage (Fig. 2a)[8,9]. HARs (2,737) and HGEs are not only distinct in the way they were defined, but they are also located in different regions of the genome. Only 7 overlapping regions were detected for HARs and HGEs identified in fetal brain (HGEs_FB, 2,104) and five such regions for HARs and HGEs identified in adult brain (HGEs_AB, 1,518). Enhancers typically exhibit a tightly regulated temporal window of activity[20], and consistent with this, HGEs are subject to dynamic regulation across development. For example, only 35 regions overlapped between HGEs_FB and HGEs_AB. As HLEs (1779) are defined by loss of enhancer marks on the human lineage, they are distinct from HGEs (8 overlapping regions between HLEs and HGEs_FB, no overlap between HLEs and HGEs_AB).

We next identified predicted target genes for HGEs and HLEs (Methods). We used previously identified target genes for HGEs_FB[11], while we leveraged new chromatin interaction profiles from the adult prefrontal cortex (PFC)[21] to identify putative target genes of HGEs_AB and HLEs, since they were defined in adult brain. We were able to assign 1518 HGEs_AB and 1779 HLEs to 1513 and 1547 putative target genes, respectively, based on chromatin interaction profiles. We first observed that the predicted target genes of HARs, HGEs, and HLEs exhibit minimal genome-wide overlaps, consistent with the notion that different classes of human evolved elements regulate different biological processes (Fig. 2b). Whereas HAR-associated genes are involved in cerebral corticogenesis and cortical lamination as described above, putative target genes for HGEs_FB are enriched for GTPase regulators and the GPCR signaling pathways[11]. In contrast, HGEs_AB interact with genes involved in collagen metabolism, TOR signaling, immune function, and lipid storage, and HLEs interact with genes involved in oxygen transport, autophagy, and thymus development (Supplementary Fig. 5a). It is particularly interesting that HGEs_AB interact with genes

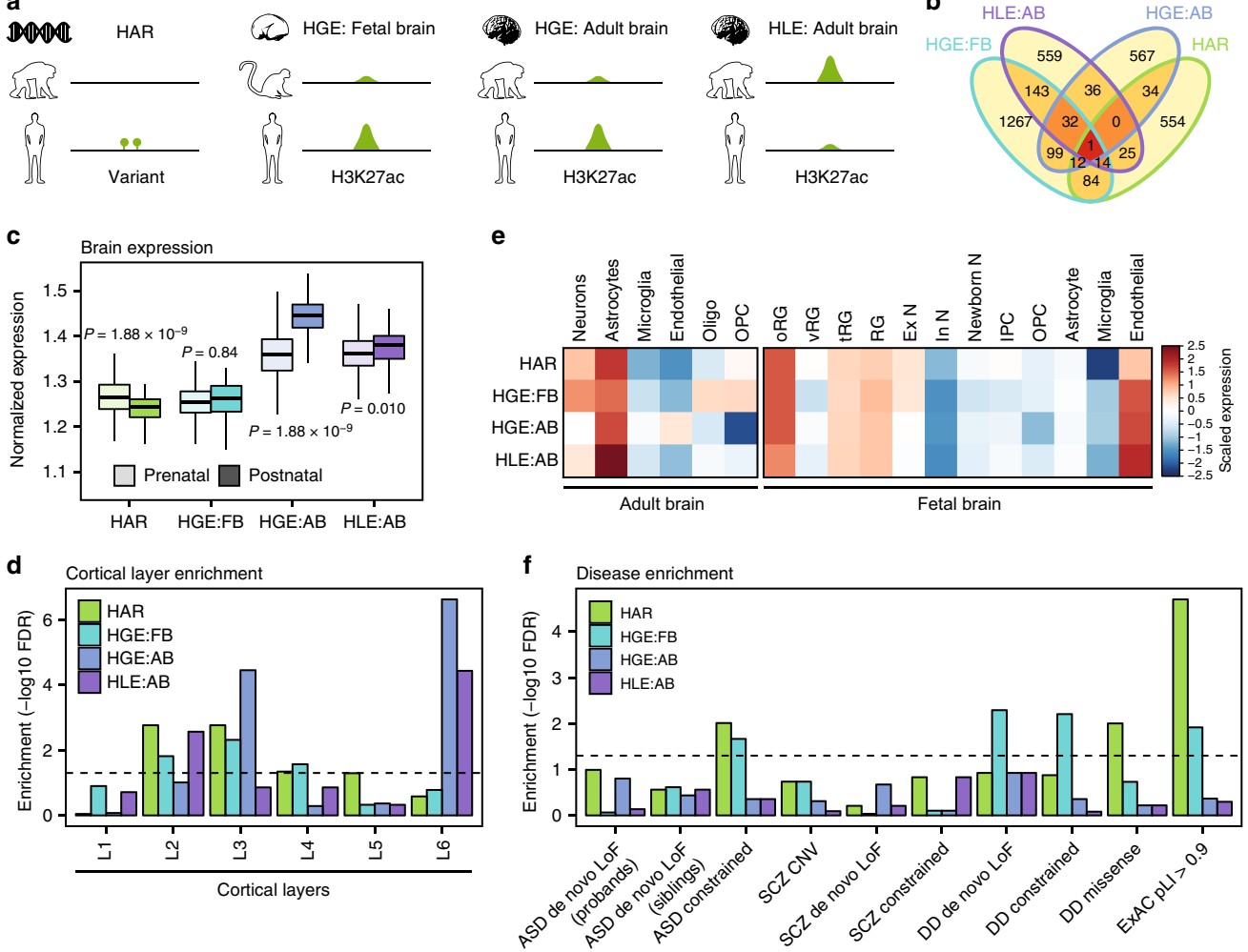

**Fig. 2** Different human evolved elements regulate distinct biological pathways. **a** Schematic of different classes of human evolved elements. HARs were defined as genomic regions with accelerated sequence changes in human;[6] HGEs_FB as the genomic regions with gained enhancer activity in human compared with rhesus macaque in developing brain;[8] HGEs_AB and HLEs as genomic regions with gained and lost enhancer activities in human compared with chimpanzee in adult brain, respectively[9]. **b** Overlap between HAR-associated genes (HAR), HGE_FB-associated genes (HGE:FB), HGE_AB-associated genes (HGE:AB), and HLE-associated genes (HLE). **c** Normalized brain expression level of genes associated with different classes of human evolved elements in prenatal and postnatal periods. FDR-adjusted P values, one-way ANOVA and post-hoc Tukey test. $n = 410$ and 453 for prenatal and postnatal samples, respectively. Center, median; box = 1st–3rd quartiles (Q); lower whisker, Q1–1.5×interquartile range (IQR); upper whisker, Q3 + 1.5×IQR. **d** Laminar specificity of genes associated with different classes of human evolved elements. **e** Human evolved element-associated genes are enriched in radial glia in the developing neocortex and astrocytes in the adult PFC. Oligo, oligodendrocytes; OPC, oligodendrocyte precursor cells; RG, radial glia; oRG, outer radial glia; vRG, ventricular radial glia; tRG, truncated radial glia; Ex N, excitatory neurons; In N, inhibitory neurons; Newborn N, newborn neurons; IPC, intermediate progenitor cells. **f** Human evolved elements interact with neurodevelopmental disorder risk genes. Also check Supplementary Fig. 5c. OR, odds ratio; ASD, autism-spectrum disorder; SCZ, schizophrenia; DD, developmental delay; LoF, loss-of-function variation; constrained, variation in LoF variation intolerant genes (pLI > 0.9);[38] pLI, probability that a gene is intolerant to LoF variation; ExAC, Exome Aggregation Consortium. P-values calculated by logistic regression correcting for coding sequence length

involved in lipid storage, as humans display an increased capacity to metabolize a lipid-rich diet, which is accompanied by the larger brain size that requires high energy demands[22].

We then explored developmental expression trajectories of putative target genes for HARs, HGEs, and HLEs. We observed distinct average expression trajectories between these groups, especially during prenatal stages, consistent with the differential enrichment of biological pathways within the target genes of each class of elements. HAR-associated genes are highly expressed during prenatal development, and are sharply upregulated during neurogenesis, peaking near mid-gestation, a period marked by neuronal migration, early neuronal phenotype definition, and dendritic arborization. HGE_FB-associated genes do not show

prenatal enrichment. HGE_AB- and HLE-associated genes are more highly expressed during postnatal development, with more pronounced postnatal enrichment for HGE_AB-associated genes (Fig. 2c). HGE- and HLE-associated genes show a pattern of more gradual upregulation throughout prenatal development, manifesting their highest expression after the peak of neuronal migration, during the period of synaptic formation and gliogenesis[23] (Supplementary Fig. 5b).

We next examined whether genes associated with human evolved elements exhibit any laminar specificity[24]. Remarkably, all classes of human evolved element-associated genes, especially HARs and HGEs_FB, were enriched in superficial cortical layers, layers 2 and/or 3, which form the inter-and intrahemispheric

connections between cortical regions and are significantly expanded in primates (Fig. 2d)[25–27]. In contrast, HGE$_{AB}$- and HLE-associated genes are also enriched in layer 6 (Fig. 2d), which projects to subcortical regions, primarily thalamus. The expansion of the superficial, supragranular layers is hypothesized to contribute to the elaboration of gyrification, as it displays the largest increase in the number of neurons and thickness in primates compared with rodents and carnivores[26,28–31]. These data therefore directly connect the expansion of hemispheric regions and their connectivity with specific molecular pathways and regulatory elements.

To further refine their functional annotation, we next determined whether human evolved element-associated genes are expressed in specific cell types by leveraging data from single-cell sequencing in the developing human neocortex and adult PFC[32,33] (Fig. 2e). In the developing cortex, all classes of human evolved element-associated genes were enriched in the outer radial glia, which comprise a major class of neural stem cells in the germinal layer that shows substantial expansion on the primate lineage[34] (Fig. 2e). This suggests that even though these different classes of human evolved elements regulate divergent biological processes, they converge on human cortical expansion, a striking finding. This observation is also consistent with the laminar patterns of enrichment described above, which highlight superficial cortical layers[30]. In the adult PFC, all classes of human evolved element-associated genes were enriched in astrocytes, while neuronal enrichment was detected for HAR- and HGE$_{FB}$-associated genes (Fig. 2e). This cell-type specificity reflects gene ontology enrichment, as HAR-associated genes are involved in neuronal proliferation and differentiation, whereas HGE$_{AB}$- and HLE-associated genes are involved in immune function. Astrocytic enrichment is particularly interesting, as human astrocytes are morphologically more complex and transcriptomically distinct from murine astrocytes[35,36], and glial co-expression networks are less preserved between rodents and humans than neuronal networks[37]. Taken together, different classes of human evolved elements potentially regulate distinct biological pathways during different developmental windows and in different cell types, although they do converge on cell types and layers responsible for cortical expansion and gyrification on the primate and human lineages.

**Human evolved elements and human-specific gene regulation.** We had previously shown that HGEs$_{FB}$ interact with protein-coding genes that are under purifying selection in primates and humans[11], so we next tested the hypothesis that protein-coding genes linked to human evolved elements are under similar selection pressures. Indeed, HAR-, HGE$_{AB}$-, and HLE-associated genes are also under purifying selection when compared with the genome background (Methods; Supplementary Fig. 6). HAR- and HGE$_{FB}$-associated genes are not only evolutionary constrained, but also enriched with genes that are intolerant to predicted loss of function (LoF) variation (pLI ≥ 0.9) in human populations[38] (Fig. 2f). In summary, human evolved genomic elements are associated with protein-coding genes that are evolutionary conserved and intolerant to haploinsufficiency, supporting the hypothesis that non-coding regulatory elements drive evolutionary divergence by species-specific transcriptional regulation of often essential, highly conserved genes[1].

As candidate genes for human evolved elements are subject to human-specific regulation, we tested whether they are differentially regulated in humans compared with non-human primates by exploiting a transcriptional atlas of human and non-human primate brain (Methods)[39,40]. Candidate genes for human evolved elements were not enriched for developmental human-specific

genes that show distinct developmental expression trajectories in human vs. rhesus macaque based on a recent study[40] (HAR, $P$ = 0.70, OR = 1.11; HGE$_{FB}$, $P$ = 0.37, OR = 1.23; HGE$_{AB}$, $P$ = 0.27, OR = 1.42; HLE, $P$ = 0.86, OR = 0.84, Fisher's exact test). In contrast, HGE$_{AB}$-associated genes showed modest, but significant, enrichment for differentially expressed genes in the adult brain tissue between humans and non-human primates (Fig. 3c, OR = 1.51, $P$ = 0.022, Fisher's exact test)[39].

Because genes associated with human evolved elements do not display distinct human-specific regulation during brain development, we hypothesized that they may be under more precise developmental control. To assess more refined developmental regulation, we first calculated the difference in relative developmental expression levels ($\Delta$ expression Z-score) at a matching developmental stage between human and rhesus macaque (Methods)[40]. We found that during prenatal and early postnatal periods (20 post-conception week (PCW) – 5 months after birth), HGE$_{FB}$-associated genes show a small, but significant increase in $\Delta$ expression Z-score (mean $\Delta$ = 0.128, $P$ = 8.2 × 10$^{-4}$, two-sided $t$-test), suggesting that HGE$_{FB}$ genes show selective expression during that stage in human brain relative to rhesus macaque brains (Fig. 3a). Genes associated with other human evolved elements do not show any deviation of $\Delta$ expression Z-scores, indicating that they do not show stage-specific enrichment in human compared with rhesus macaque (Supplementary Fig. 7).

Another characteristic of human-specific transcriptional regulation is the observation of early breakpoints during brain development, which denote a group of genes that display abrupt expression changes in human compared to rhesus macaque[40]. Genes with early breakpoints are thought to represent an earlier onset of developmental processes that are potentially extended in human brain compared with non-human primates[40]. Notably, HAR-associated genes tend to have earlier breakpoints (Fig. 3b, Methods), implying that HARs may contribute to the expansion and elaboration of human cortex by inducing early peaks and more protracted expression of essential regulators of neuronal proliferation and differentiation. HAR-associated genes with earlier breakpoints include *CPLX2*, whose protein product functions in synaptic vesicle exocytosis[41] and *ITPR1*, which harbors mutations found in spinocerebellar ataxia[42]. In contrast, HGE$_{AB}$-associated genes show delayed breakpoints (Fig. 3b), suggesting that human evolved elements active in adult brain regulate genes with later peak expression during development. These genes include *S100B*, a marker for astrocytes[43]. HAR$_{FB}$- and HLE-associated genes do not exhibit significant changes in breakpoints in the genes they regulate (Supplementary Fig. 7), indicating that the activity of different classes of human evolved regulatory elements manifest distinct developmental trajectories.

We also hypothesized that human evolved elements might mediate human-specific gene co-regulation. To address this question, we leveraged recently identified human-specific co-expression modules[44] to gain further insights into human-specific gene regulation mediated by human evolved elements. Indeed, HGE$_{FB}$- and HGE$_{AB}$-associated genes were enriched in human-specific modules, M162 (Methods, OR = 3.12, $P$ = 2.06 × 10$^{-4}$, Fisher's exact test) and M122 (OR = 7.12, $P$ = 3.45 × 10$^{-5}$, Fisher's exact test), respectively (Fig. 3c). Genes in M162 are associated with alternative splicing and expressed in a specific subgroup of excitatory neurons, while M122 is not associated with a specific gene ontology or cell type[44]. Given that alternative splicing is hypothesized to play an essential role in transcriptomic complexity and diversity and subject to dynamic regulation in humans compared with non-human primates[45], it is of note that HGE$_{FB}$-associated genes are differentially regulated in human and associated with alternative splicing in a subset of excitatory neurons.

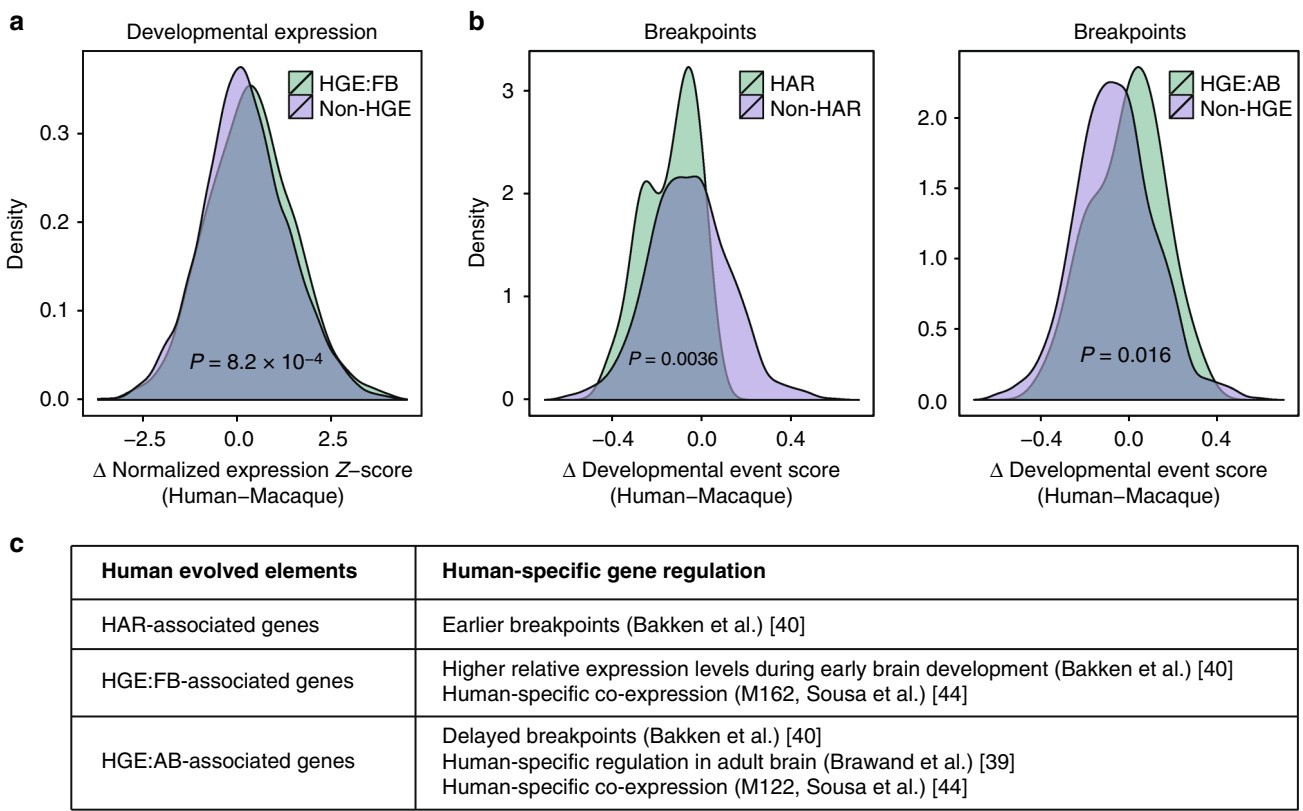

**Fig. 3** Human evolved elements mediate human-specific regulation. **a** HGE$_{FB}$-associated genes show relative higher expression during early brain development in human compared with rhesus macaque. *P*-values calculated by two-sided *t*-tests. **b** HAR-associated genes show earlier breakpoints, while HGE$_{AB}$-associated genes show later breakpoints. *P*-values calculated by two-sided Wilcoxon rank sum tests. **c** Distinct classes of human-evolved elements are associated with different types of human-specific gene regulation

**Human cortical evolution and neurodevelopmental disorders.** We next hypothesized that regulatory elements that drive human brain evolution may affect susceptibility to neurodevelopmental disorders via their target genes because (1) LoF-intolerant genes are enriched for de novo LoF variation in ASD and developmental delay (DD)[46], (2) HARs are enriched with biallelic mutations in consanguineous ASD families[10], and (3) HGEs$_{FB}$ interact with genes associated with intellectual disability[11]. Indeed, we observed that HAR- and HGE$_{FB}$-associated genes are enriched with LoF-intolerant genes that harbor de novo mutations in ASD (ASD constrained genes) and DD risk genes[47] (Fig. 2f, Supplementary Fig. 5c). In contrast, HGE$_{AB}$- and HLE-associated genes are enriched with genes affected by copy number variation (CNV) in schizophrenia[48], an adolescent- and adult-onset disorder. It is also interesting to note that although both HGE and HAR regulated genes are implicated in ASD, the overall patterns predict slightly different relationships to neurodevelopmental disease. Relative to HGE$_{FB}$, genes putatively regulated by HARs show more enrichment in ASD constrained genes and de novo LOF variation, whereas HGE$_{FB}$ appear more enriched for constrained DD genes, or genes harboring LOF mutations in DD (Supplementary Fig. 5c).

**Functional validation of HAR-associated genes.** Although chromatin contacts provide a powerful tool to identify long-range physical chromatin interactions necessary for gene regulation, experimental validation would increase confidence that these chromatin contacts were functional. We therefore experimentally validated the functional impact of a subset of HARs active in developing human brain using primary human neural progenitor cells (phNPCs), which are a well validated in vitro model system for human neural development. We chose 3 enhancer-gene predictions based on the known role of the target gene in neurodevelopment, *GLI2, GLI3,* and *TBR1* and present all of the results for these predicted interactions. We targeted catalytically inactive Cas9 linked to the synthetic VP64 activation domain (dCas9-VP64) to three HARs in phNPCs whose putative target genes include *GLI2*, the promoter of which interacts with HAR-01246 ~330 kb away (Fig. 4a). *GLI2* encodes a C2H2-type zinc-finger protein that mediates Sonic hedgehog (Shh) signaling and is critical for the induction of neural tube in mice[49,50]. Targeting dCas9-VP64 to the HAR using two guide RNAs (gRNAs) resulted in a ~60% increase in the expression level of *GLI2* in phNPCs (Fig. 4a). CRISPR/Cas9-mediated transcriptional activation of HAR-02296 that interacts with *GLI3* also led to a 30–40% increase in its expression (Fig. 4b). GLI3 is required for dorsal–ventral patterning of telencephalon including the formation of the cortical hem in mouse and humans[51,52] and *Gli3* null mice display a substantially smaller neocortex and absence of the hippocampus[53]. *TBR1*, a marker for deep layer projection neurons in the developing cortex that specifies laminar identity of cortical neurons[54,55], interacts with HAR-01298, which is ~170 kb distal (Fig. 4c). Due to the small size of HAR-01298 (16 bp), we could not find gRNAs directly targeting this element, so instead we designed gRNAs flanking the region. One gRNA targeting the region 88 bp upstream of HAR-01298 increased *TBR1* expression up to 80%, while the other gRNA targeting the region 92 bp downstream of HAR-01298 did not affect TBR1 expression (Fig. 4c).

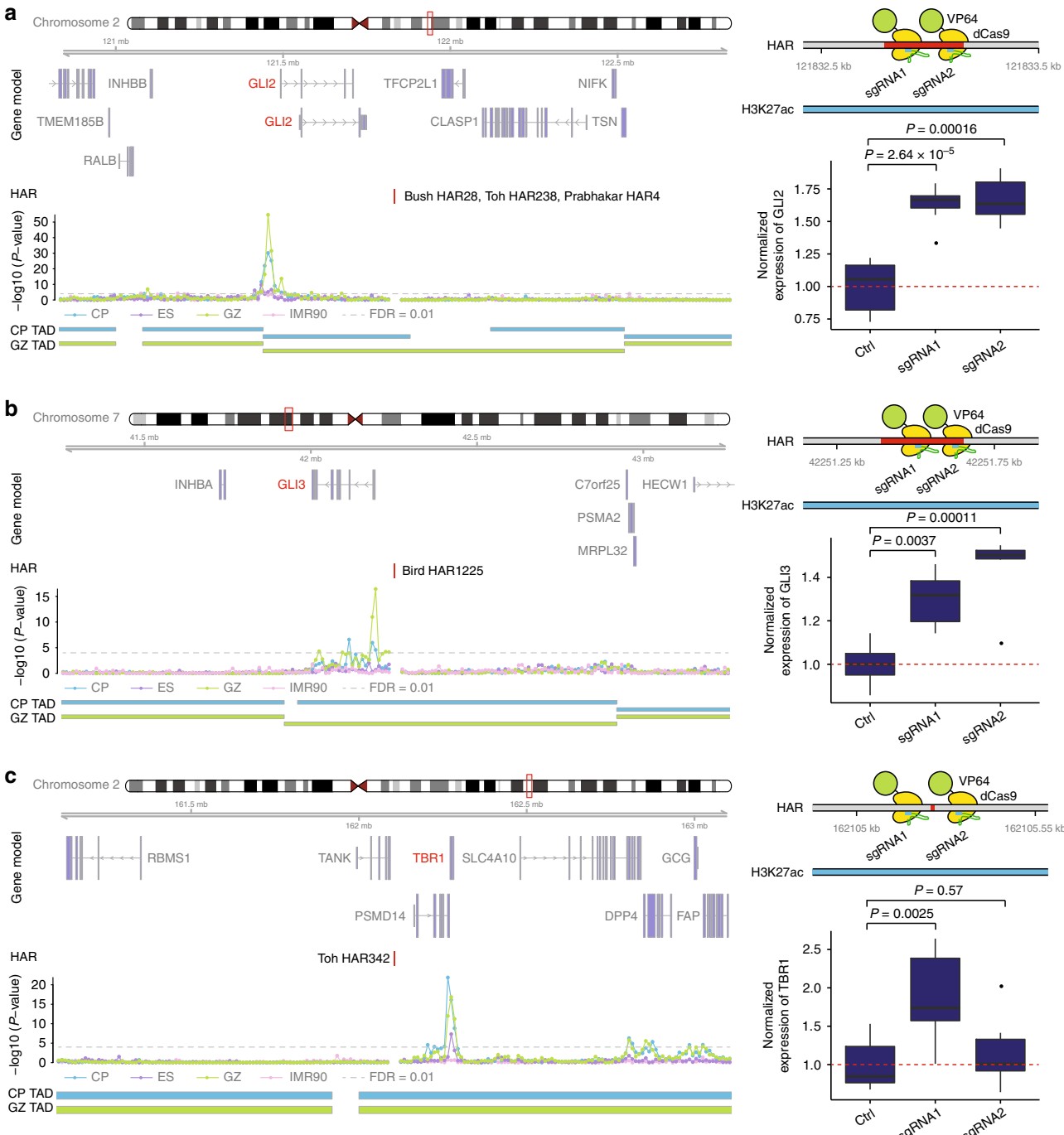

**Fig. 4** Functional validation of HAR-interacting genes. **a–c** Left, chromatin interaction map of HARs that interact with *GLI2* (**a**), *GLI3* (**b**), and *TBR1* (**c**). Gene Model is based on Gencode v19 and possible target genes are marked in red; Genomic coordinate for HARs is labeled as HAR; −log10 (*P*-value), significance of the interaction between HARs and each 10 kb bin, gray dotted line for FDR = 0.01; TAD borders in CP and GZ are shown. Right, targeted binding sites for two guide RNAs (gRNAs). The HAR is located in the active enhancer marks (H3K27ac) in fetal brain. Targeting dCas9-VP64 to HARs in primary human neural progenitor cells (phNPCs) results in an 30–80% increase in the expression level of putative target genes predicted by Hi-C. Normalized expression levels of putative target genes (*GLI2* (**a**), *GLI3* (**b**), and *TBR1* (**c**)) relative to control (Ctrl). n = 6 (Ctrl), 8 (gRNA1), 6 (gRNA2) for *GLI2*; 7 (Ctrl), 6 (gRNA1 and gRNA2) for *GLI3*; 5 (Ctrl), 6 (gRNA1 and gRNA2) for *TBR1*. *P*-values, one-way ANOVA and post hoc Tukey test. Center, median; box = 1st–3rd quartiles (Q); lower whisker, Q1 − 1.5×interquartile range (IQR); upper whisker, Q3 + 1.5 × IQR

## Discussion

We leveraged chromatin architecture in fetal and adult human cerebral cortex[11,21] to identify the putative regulatory targets of non-coding elements that have been previously identified as those most changing on the human lineage[2–9]. By comparing the two major different classes of human evolved elements, those based on

sequence changes, and those based on functional epigenetic alterations, we found that multiple modalities of regulatory relationships likely drive human brain evolution by orchestrating different molecular programs in distinct developmental windows and cell types. For example, HAR-associated genes are prenatally enriched, HGE$_{AB}$- and HLE-associated genes are postnatally enriched,

while HGE$_{FB}$-associated genes do not show developmental-stage specific enrichment and are expressed across development as a group. In postnatal brain, HAR target genes are predominantly expressed in excitatory neurons, while HGE- and HLE-associated genes are expressed in astrocytes and neural stem cells.

Critically, despite the observation that different human evolved elements are predicted to modulate distinct biological pathways, there are areas of convergence. In developing brain, genes regulated by both HGE and HAR converge on radial glia, a major neurogenic niche in developing human cortex[56]. In adult brain, they converge on the supragranular layers, which are most expanded in primates, especially humans, and mediate connections between different cortical regions, as well as between the two cerebral hemispheres[25]. The expansion of supragranular layers in human is attributed to the enlargement of outer subventricular zone (OSVZ), the layer in which the majority of human radial glia are located[34,57], suggesting that human specific gene regulation converges on human cortical expansion and its subsequent intra- and inter-cortical connectivity, which is the major anatomical feature of human brain evolution[26]. Since it is gene regulation, rather than changes in protein coding sequence that is the major distinguishing feature between humans and non-human primates[1], these data provide a fundamental molecular map linking human specific gene regulation to brain evolution.

Human evolved elements do not only differ in the biological processes they regulate, but the types of human-specific regulation to which they are subject. HAR-associated genes show earlier onset of expression that continues throughout brain development, while HGE$_{AB}$-associated genes show later onset of expression in human compared with rhesus macaque. On the contrary, HGE$_{FB}$-associated genes show higher relative prenatal and early postnatal expression in human than rhesus macaque. This observation suggests that multiple forms of gene regulation contribute to the evolution of uniquely human traits.

The functional differentiation between elements whose evolution is based on sequence vs. those based on epigenetic changes also extends to their relationship to human brain disorders. Genes regulated by human evolved elements in developing brain (HARs and HGEs$_{FB}$) are both associated with neurodevelopmental disorders. However, HAR genes appear more likely to be disrupted in ASD, while HGE$_{FB}$ genes are more substantially enriched in constrained or LOF DD genes although enriched in constrained ASD genes (Supplementary Fig. 5). In contrast to elements active in fetal brain[58], human evolved elements in adult brain (HGEs$_{AB}$ and HLEs) are associated with later onset psychiatric disorders, suggesting that the susceptibility to neuropsychiatric disorders is related, at least partially, to human-specific developmental gene regulation. These data not only identify the genomic elements and their target genes that underlie these human specific features, but also explain in part why behavioral outcome measures in rodent models may not be directly translatable to human disease in many cases.

We employed CRISPR/Cas9-mediated transcriptional activation system to functionally validate the effects of HARs on putative target genes (GLI2, GLI3, and TBR1) that encode essential regulators of forebrain development and cortical lamination. Both GLI2 and GLI3 are involved in patterning and growth of the central nervous system (CNS) regulated by Sonic Hedgehog (SHH). Knockdown of Gli2 in neuroepithelial cells inhibits the expression of neural stem cell markers and induces premature differentiation of neural stem cells[59], whereas Gli3 hypomorphic mutant mice display perturbed cortical lamination[60]. Further, single-cell transcriptomic profiles on developing human cortex demonstrated that both GLI2 and GLI3 are enriched in radial glia, where GLI2 is specifically enriched in young outer radial glia, and GLI3 is an essential component of gene

expression cascades in early cortical neurogenesis[32]. Therefore, HAR-02296 and HAR-01246 may be involved in human brain evolution by regulating cortical expansion and lamination. Another example is TBR1, a well-known marker for deep layer neurons. Rostral markers are substantially downregulated in Tbr1 null mice, highlighting its potential role in establishing frontal cortex identity[54]. Moreover, recurrent de novo loss-of-function (LoF) variants in TBR1 have been identified in individuals with ASD[61,62], and TBR1 itself regulates other ASD risk genes[63]. Thus, HAR-01298, which we show regulates TBR1 expression may coordinate patterning of the frontal cortex, the disruption of which can lead to neurodevelopmental disorders. Notably, HARs and HGE$_{FB}$ also interact with genes that regulate the size of the frontal cortex such as FGF17 and EMX2[64], a functional link between human-evolved elements and evolutionary expansion of the frontal cortex in human[65].

Collectively, these findings illustrate how changes in gene regulation mediated by rapid evolution of non-coding regions contribute to phenotypic differences between human and non-human primates despite the high degree of similarities and high levels of constraint in protein-coding sequences. These data are consistent with a model whereby newly evolved biological mechanisms driving human cerebral cortical evolution increase vulnerability for a range of neuropsychiatric and neurodevelopmental conditions. Further, they provide a framework that links regulatory elements to target genes that will be of substantial utility for mechanistic studies and disease modeling.

## Methods

**Enrichment of HARs in regulatory elements.** We employed GREAT[66] to analyze chromatin states/histone marks enrichment for HARs. We calculated the proportion of a chromatin state over the genome ($p$), the number of HARs ($n$), and the number of HARs that overlap with a given chromatin state. The significance of enrichment was calculated by the binomial probability of $P = Pr_{binom}$ ($k \geq s/n = n$, $p = p$). Fold enrichment was calculated as the ratio between (the proportion of HARs in the genome that overlap with a chromatin state) and (the proportion of HARs in the genome × the proportion of chromatin state in the genome).

Because HARs are evolutionary conserved elements, evolutionary conservation can affect enrichment patterns in regulatory elements. We conducted a secondary enrichment analysis controlling for evolutionary conserved elements. We defined evolutionary conserved regions as genomic regions larger than 20 bp with a phastCons score >0.40 (http://compgen.cshl.edu/phast/, R library phastCons100way.UCSC.hg19). Then, we performed a Fisher's exact test with the following contingency table (Table 1).

In addition, we randomly selected 10,000 sets of genomic regions that have matched size and phastCons scores with HARs (hereby referred as default regions). We then overlapped these default regions to DNase I hypersensitivity sites (DHS) in each tissue/cell type and obtained an odds ratio (OR) for each Fisher's exact test (same contingency table used as above). This leads to a set of 10,000 ORs for each tissue type, which was plotted in Supplementary Fig. 1a. Collectively, we used three metrics (GREAT enrichment, Fisher's test while controlling for evolutionary conservation, 10,000 permutations) to confirm that HARs show highest enrichment for DHS in the fetal brain compared with other tissue/cell types.

DHS in multiple cell/tissue types and 15-chromatin states (Table 2) in fetal brain and adult prefrontal cortex (PFC) were obtained from Roadmap Epigenome[12], and chromatin accessibility peaks in cortical plates (CP) and germinal zone (GZ) were obtained from de la Torre-Ubieta et al.[13].

We detected strong enrichment signals for HARs in regulatory elements of the developing brain, but not in the adult cortex. However, it is difficult to distinguish the effects of the development from the effects of the cellular heterogeneity and/or regional differences. This is because (1) tissue-level DHS lack cellular resolution, and (2) fetal brain DHS lack specific regional coordinates[12]. To address this issue, we leveraged ATAC-seq peaks obtained from two cortical layers with well-established cellular identities (CP is comprised of post-mitotic neurons, while GZ is comprised of neural progenitors) and regional coordinates (paracentral cortex)[13]. We were able to detect robust enrichment for HARs in ATAC-seq peaks in fetal cortex, implicating strong developmental effects. However, given that accessible chromatin in GZ was more enriched for HARs than accessible chromatin in CP, we believe cellular heterogeneity also plays a role. The distinction will become clearer once more cellularly, regionally defined epigenomic landscape becomes available.

**Identification of the putative target genes of HARs.** HARs were categorized into (1) coding HARs that reside in exons, 5′ untranslated regions (UTR), 3′ UTR, promoters (1 kb upstream to transcription start sites), and downstream flanking

**Table 1 Contingency table for calculating cell-type specific enrichment of HARs**

| | DHS | Not DHS | Row total |
|---|---|---|---|
| HAR | # of HARs that overlap with DHS in a given cell type $= A$ | # of HARs that do not overlap with DHS in a given cell type $= D$ | $A + C$ |
| Background (evolutionary conserved regions) | # of evolutionary conserved regions that overlap with DHs in a given cell type $= B$ | # of evolutionary conserved regions that do not overlap with DHS in a given cell type $= D$ | $B + D$ |
| Column total | $A + B$ | $C + D$ | $N ( = A + B + C + D)$ |

**Table 2 Annotations for chromatin states**

| | |
|---|---|
| TssA | Active Transcription start sites (TSS) |
| TssAFlnk | Flanking active TSS |
| TxFlnk | Transcription at gene 5′ and 3′ |
| Tx | Strong transcription |
| TxWk | Weak transcription |
| EnhG | Genic enhancers |
| Enh | Enhancers |
| ZNF/Rpts | ZNF genes & repeats |
| Het | Heterochromatin |
| TSSBiv | Bivalent/Poised TSS |
| BivFlnk | Flanking Bivalent TSS/Enhancers |
| EnhBiv | Bivalent Enhancer |
| ReprPC | Repressed PolyComb |
| ReprPCWk | Weak Repressed PolyComb |
| Quies | Quiescent |

regions (1 kb downstream to transcription end sites), and (2) non-coding HARs that reside in intergenic and intronic regions. Coding HARs were directly assigned to their target genes based on their genomic coordinates, while non-coding HARs were annotated based on chromatin interactions[11]. As the highest resolution available for Hi-C data was 10 kb, we assigned non-coding HARs to 10 kb bins, and obtained Hi-C interaction profiles of the 1 Mb flanking regions for each HAR-containing bin.

We also obtained background Hi-C interaction profiles from randomly selected genomic regions that share similar properties with HARs. For each HAR, we randomly selected a region within the same chromosome that has the same length and GC content (<5% difference) with the HAR. We repeated this 40 times to construct $2737 \times 40 = 109,480$ randomly matched regions. As some of these regions were overlapping, we ended up having 109,408 randomly selected genomic regions with matched GC content and length as HARs (Supplementary Table 4), which we used to construct a null distribution. Using these background Hi-C interaction profiles, we fit the distribution of Hi-C contacts at each distance for each chromosome using the Weibull distribution in *fitdistrplus* package. Significance for a given Hi-C contact was then calculated as the probability of observing a stronger contact under the null distribution matched by chromosome and distance. *P*-values were adjusted to the number of HARs (non-coding, 2634) and bins (198 bins per locus), and Hi-C contacts with FDR < 0.01 were selected as significant interactions. Putative target genes were identified by overlapping HAR interacting regions with promoter coordinates (2 kb upstream to TSS, Gencode v19). The same analysis was performed on Hi-C interaction profiles in CP, GZ, embryonic stem (ES) cells, and IMR90 cells[11].

**Identification of the putative target genes of HGEs and HLEs.** HGEs$_{AB}$ and HLEs were defined as genomic regions that underwent regulatory changes between human and chimpanzee in adult brain[9], whereas HGEs$_{FB}$ were defined as genomic regions that underwent regulatory changes between human and rhesus macaque in developing cortex[8]. To assign HGEs$_{FB}$ to their target genes, we used previously identified target genes of HGEs$_{FB}$[11] with a slight modification. We first categorized HGEs$_{FB}$ into ones located in promoters and ones that are not. Promoter HGEs$_{FB}$ were directly assigned to their target genes based on their genomic coordinates, while non-promoter HGEs$_{FB}$ were assigned to their target genes based on chromatin interaction profiles in fetal brain. We only used non-promoter HGEs$_{AB}$ and HLEs for target gene assignment. To assign HGEs$_{AB}$ and HLEs to their target genes, we first converted genomic coordinates of HGEs$_{AB}$ and HLEs from hg38 to hg19 using liftOver (https://genome.ucsc.edu/cgi-bin/hgLiftOver). HGEs$_{AB}$ and HLEs were then assigned to their target genes using newly generated chromatin interaction profiles in the adult PFC[21]. As HGEs and HLEs were respectively defined by gain and loss of H3K27ac marks, we only used promoter-based interactions.

**Gene enrichment analysis and cell-type enrichment analysis.** Gene ontology (GO) enrichment was performed by GO-Elite Pathway Analysis (EnsMart77, http://www.genmapp.org/go_elite/). Genes that reside within 1 Mb flanking regions from each human evolved element were used as a background gene list.

Gene lists for disease enrichment analysis were curated from several sources: genes that harbor de novo loss-of-function (LoF) variation in ASD were obtained from Iossifov et al.[61] and de Rubeis et al.[62]; genes that harbor de novo LoF variation in schizophrenia and developmental delay (DD) were obtained from Fromer et al.[67] and the Deciphering Developmental Disorders Study[68], respectively; constrained genes associated with ASD, schizophrenia, and DD were obtained from Kosmicki et al.[46]; pathogenic missense variants for ASD and DD were obtained from Samocha et al.[47]; and genes intolerant for LoF variation were obtained from The Exome Aggregation Consortium (ExAC)[38]. Genes that reside within copy number variants (CNV) in schizophrenia were obtained from Marshall et al.[48] Human-specific genes were obtained from Bakken et al.[40] (Table S10), Sousa et al.[44] (Table S5), and Brawand et al.[39] (Table S1).

To calculate enrichment statistics, we used logistic regression in R:
*glm.out<- glm(evol.list~gene.list+covariates, family=binomial)*
*P.value<- summary(glm.out)$coefficients[2,4]*
We built two vectors (*evol.list* and *gene.list*) of 0 or 1 with a length of background gene lists. A vector *evol.list* denotes for human-evolved element associated genes, while *gene.list* is a vector for a curated gene list (0 = not included in the gene list, 1 = included in the gene list).

For disease enrichment analysis, we used all protein-coding genes based on biomaRt (Gencode v19) as a background gene list (19,154 genes), since de novo mutations were identified from exome studies that only contain protein-coding genes. Since de novo mutation rates are dependent on the coding exon length, we regressed out exon length by adding *covariates* = exon length for a given gene.

We also performed enrichment analysis with human-specific genes and co-expression modules reported by Bakken et al.[40], Sousa et al.[44], and Brawand et al.[39] For example, Bakken et al. reported 197 genes that show human-specific developmental expression patterns. As Bakken et al. used a microarray-based platform in rhesus macaque, we used 10,715 genes where human orthologs were available as a background gene list. Sousa et al.[44] used RNA-seq, and we used 19,154 protein-coding genes as a background list. Moreover, we leveraged brain expression data (total 22 samples: 6 human vs. 16 primates including Gorilla, Pan Trogodytes, Pongo Pygmaeus, and Rhesus Macaque) from Brawand et al.[39] to run differential expression analyses between human and primates using *lm (expression~species + sex)*. In total, 590 genes were differentially expressed in human vs. primates at an FDR<0.05. We used 13,080 primate orthologs as a background gene list. For cell-type enrichment analysis and enrichment analysis with human-specific genes, we did not add covariates in the logistic regression. The test becomes equivalent to Fisher's exact test. After calculating the enrichment *P*-values, we performed a multiple correction by counting for both curated gene sets and classes of evolutionary elements.

Since HARs are evolutionary conserved elements and HGEs are enhancers, the enrichment with neurodevelopmental disorder risk genes could be simply due to their genomic features (i.e. evolutionary conservation and being a regulatory element). To further confirm that this enrichment is not merely due to their genomic features, we performed a disease enrichment analysis for HAR-associated genes compared with other genes associated with evolutionary conserved elements (3,083,588 evolutionary conserved elements with phastCons score >0.40 (similar to most HARs) were mapped to 16,676 protein-coding genes), and HGE$_{FB}$-associated genes over genes associated with fetal brain enhancers (186,304 enhancers reported in Reilly et al.[8] were mapped to total 15,693 protein-coding genes), where we obtained similar results (Supplementary Fig. 5c).

**Developmental and cellular expression profiles.** The spatiotemporal transcriptomic atlas data from human brain was obtained from Kang et al.[69] As this dataset contains expression values from multiple brain regions, we selected transcriptomic profiles of cerebral cortex with developmental epochs that span prenatal (6–37 post-conception weeks, PCW) and postnatal (4 months–42 years) periods. Expression values were log-transformed and centered to the mean expression level for each sample using a *scale(center = T, scale = F) + 1* function in R. Genes associated with human evolved elements were selected for each sample and their average centered expression values were calculated and plotted.

Cell-type specific expression profiles in the adult PFC and developing neocortex were obtained from Darmanis et al.[33] and Nowakowski et al.[32], respectively. We processed single-cell expression values by centering to the mean expression level for each cell using a $scale(center = T, scale = F)$ function in R. This results in centered expression values denoting each gene's relative expression level in a given cell, referred as cell-level centered expression values. We then calculated average cell-level centered expression values of genes mapped to each class of human-evolved elements.

**Evolutionary constraints on protein-coding genes.** Protein-coding genes (Gencode v19) were selected and the ratio between non-synonymous substitution (dN) and synonymous substitution (dS) was calculated by biomaRt for homologs in mouse, rhesus macaque, and chimpanzee for representation of mammals, primates, and great apes, respectively. The log2(dN/dS) distribution for protein-coding genes that interact with each class of human evolved elements was plotted against distribution for protein-coding genes not associated with human evolved elements. If log2(dN/dS) < 0 for a protein-coding gene, it indicates that the gene is under purifying selection. If log2 (dN/dS) > 0, the gene is under positive selection.

**Evolutionary transcriptional regulation.** Transcriptional maps for 4125 orthologous genes in rhesus macaque and human brain were obtained from Bakken et al.[40] Expression values of each gene were normalized to developmental time points using the *scale* function in R to calculate normalized expression Z-scores. Thus, normalized expression Z-scores denote relative developmental expression enrichment at a given developmental epoch. For example, if the Z-score of a gene A is high at post-conception week (PCW) 20, it means that the gene is highly expressed in that developmental stage compared with other developmental stages. We then subtracted normalized expression Z-scores at available matching developmental time points (based on developmental event scores[40], Table 3) between human and rhesus. A positive Δ expression Z-score indicates that the gene is more enriched at a given developmental epoch in human than in rhesus. We then compared the distribution of Δ expression Z-scores for HAR-, HGE-, and HLE-associated genes with non-HAR-, non-HGE-, and non-HLE-associated genes, respectively.

We used the difference in breakpoints between human and rhesus calculated by Bakken et al.[40] Timescales of breakpoints were calculated based on developmental event scores. Negative breakpoints denote early breakpoints in human compared with rhesus. We then compared the distribution of Δ breakpoints for HAR-, HGE-, and HLE-associated genes with non-HAR-, non-HGE-, and non-HLE-associated genes, respectively.

We also leveraged recently published species-specific weighted gene co-expression correlation network analysis (WGCNA) modules[44] to test whether genes associated with human evolved elements are enriched in co-expression networks with human-specific expression signatures. Logistic linear regression with a background list of genes included in the WGCNA analysis[44] was used to calculate the significance of enrichment. We did not regress out exome or gene length, as promoter-based interactions were used.

**CRISPR/Cas9-mediated transcriptional activation of HARs.** To experimentally validate Hi-C predicted candidate target genes of HARs, we chose HARs that (1) overlap with H3K27ac marks in fetal brain[8] and (2) interact with developmentally important genes expressed in human neural progenitors (Supplementary Fig. 8), and that are 3) predicted to regulate only one gene. These criteria identified three HARs (HAR-01246, HAR-02296, HAR-01298) that interact with *GLI2*, *GLI3*, and *TBR1*, respectively. Two sets of guide RNAs (gRNAs) targeting different regions of HARs that interact with *GLI2*, *GLI3*, and *TBR1* were designed by benchling (https://benchling.com/). As HAR-01298 is 16 bp in size, we could not design gRNAs targeting such a small region. Therefore, we targeted two gRNAs flanking this HAR. These gRNAs were cloned into an EF1a-dCas9-VP64-2A-GFP-sgRNA vector (modified from Addgene, 61422). An empty vector without any gRNA insertion was used as control. Virus was generated by co-transfection of CRISPR vectors with pVSVg (Addgene, 8454) and psPAX2 (Addgene, 12260) in HEK293 cells. Primary human neural progenitor cells (phNPC) were infected with viruses (empty vectors, gRNA1, gRNA2 for each HAR) on the day of split and differentiated with Neurobasal A (Invitrogen) supplemented with B27 (Gibco), Gluta-MAX (Gibco), antibiotics and antimycotics (Gibco), BDNF (10 ng/mL; Peprotech), and NT-3 (10 ng/mL; Peprotech). Half of the media was replaced three times per week during the differentiation (also see ref. [11]). After 2.5 weeks of differentiation,

cells that were infected (GFP+) were sorted by FACS. RNA was extracted by miRNeasy Mini Kit (Qiagen) and the expression level of putative target genes (*GLI2*, *GLI3*, and *TBR1*) was measured by qPCR (LightCycler 480 SYBR Green I Master, Roche) and normalized to *GAPDH*. gRNA and primer sequences for both genomic DNA and qPCR are described in Supplementary Table 3.

**Reporting summary.** Further information on research design is available in the Nature Research Reporting Summary linked to this article.

## Data availability

Hi-C data from fetal brain are available through GEO and dbGaP under the accession number GSE77565 and phs001190.v1.p1, respectively. Hi-C data from adult brain are available through https://www.synapse.org//#!Synapse:syn4921369/wiki/390671 and the PsychENCODE knowledge portal http://resource.psychencode.org/. Promoter-based interaction maps for fetal brain and adult brain are available in Supplementary Tables 22–23 of Won et al.[11] and the PsychENCODE knowledge portal, respectively.

## Code availability

Codes used to analyze and plot the results are available in the Supplementary Software 1.

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

---

### Table 3 Matching developmental event scores and corresponding developmental time points

| Human | Rhesus macaque |
| --- | --- |
| 0.46 (PCW 20) | 0.48 |
| 0.54 (PCW 26) | 0.51 |
| 0.76 (5 months after birth) | 0.77 |

24. He, Z. et al. Comprehensive transcriptome analysis of neocortical layers in humans, chimpanzees and macaques. *Nat. Neurosci.* **20**, 886–895 (2017).
25. Molyneaux, B. J., Arlotta, P., Menezes, J. R. & Macklis, J. D. Neuronal subtype specification in the cerebral cortex. *Nat. Rev. Neurosci.* **8**, 427–437 (2007).
26. Geschwind, D. H. & Rakic, P. Cortical evolution: judge the brain by its cover. *Neuron* **80**, 633–647 (2013).
27. Bystron, I., Blakemore, C. & Rakic, P. Development of the human cerebral cortex: Boulder Committee revisited. *Nat. Rev. Neurosci.* **9**, 110–122 (2008).
28. Hutsler, J. J., Lee, D. G. & Porter, K. K. Comparative analysis of cortical layering and supragranular layer enlargement in rodent carnivore and primate species. *Brain Res.* **1052**, 71–81 (2005).
29. Richman, D. P., Stewart, R. M., Hutchinson, J. W. & Caviness, V. S. Jr. Mechanical model of brain convolutional development. *Science* **189**, 18–21 (1975).
30. Kennedy, H. & Dehay, C. Self-organization and interareal networks in the primate cortex. *Prog. Brain Res.* **195**, 341–360 (2012).
31. Rakic, P. A small step for the cell, a giant leap for mankind: a hypothesis of neocortical expansion during evolution. *Trends Neurosci.* **18**, 383–388 (1995).
32. Nowakowski, T. J. et al. Spatiotemporal gene expression trajectories reveal developmental hierarchies of the human cortex. *Science* **358**, 1318–1323 (2017).
33. Darmanis, S. et al. A survey of human brain transcriptome diversity at the single cell level. *Proc. Natl Acad. Sci. USA* **112**, 7285–7290 (2015).
34. LaMonica, B. E., Lui, J. H., Wang, X. & Kriegstein, A. R. OSVZ progenitors in the human cortex: an updated perspective on neurodevelopmental disease. *Curr. Opin. Neurobiol.* **22**, 747–753 (2012).
35. Zhang, Y. et al. Purification and characterization of progenitor and mature human astrocytes reveals transcriptional and functional differences with mouse. *Neuron* **89**, 37–53 (2016).
36. Oberheim, N. A., Wang, X., Goldman, S. & Nedergaard, M. Astrocytic complexity distinguishes the human brain. *Trends Neurosci.* **29**, 547–553 (2006).
37. Miller, J. A., Horvath, S. & Geschwind, D. H. Divergence of human and mouse brain transcriptome highlights Alzheimer disease pathways. *Proc. Natl Acad. Sci. USA* **107**, 12698–12703 (2010).
38. Lek, M. et al. Analysis of protein-coding genetic variation in 60,706 humans. *Nature* **536**, 285–291 (2016).
39. Brawand, D. et al. The evolution of gene expression levels in mammalian organs. *Nature* **478**, 343–348 (2011).
40. Bakken, T. E. et al. A comprehensive transcriptional map of primate brain development. *Nature* **535**, 367–375 (2016).
41. McMahon, H. T., Missler, M., Li, C. & Südhof, T. C. Complexins: cytosolic proteins that regulate SNAP receptor function. *Cell* **83**, 111–119 (1995).
42. Zambonin, J. L. et al. Spinocerebellar ataxia type 29 due to mutations in ITPR1: a case series and review of this emerging congenital ataxia. *Orphanet J. Rare Dis.* **12**, 121 (2017).
43. Raponi, E. et al. S100B expression defines a state in which GFAP-expressing cells lose their neural stem cell potential and acquire a more mature developmental stage. *Glia* **55**, 165–177 (2007).
44. Sousa, A. M. M. et al. Molecular and cellular reorganization of neural circuits in the human lineage. *Science* **358**, 1027–1032 (2017).
45. Calarco, J. A. et al. Global analysis of alternative splicing differences between humans and chimpanzees. *Genes Dev.* **21**, 2963–2975 (2007).
46. Kosmicki, J. A. et al. Refining the role of de novo protein-truncating variants in neurodevelopmental disorders by using population reference samples. *Nat. Genet.* **49**, 504–510 (2017).
47. Samocha, K. E. et al. Regional missense constraint improves variant deleteriousness prediction. *bioRxiv* https://doi.org/10.1101/148353 (2017).
48. Marshall, C. R. et al. Contribution of copy number variants to schizophrenia from a genome-wide study of 41,321 subjects. *Nat. Genet.* **49**, 27–35 (2017).
49. Ding, Q. et al. Diminished Sonic hedgehog signaling and lack of floor plate differentiation in Gli2 mutant mice. *Development* **125**, 2533–2543 (1998).
50. Matise, M. P., Epstein, D. J., Park, H. L., Platt, K. A. & Joyner, A. L. Gli2 is required for induction of floor plate and adjacent cells, but not most ventral neurons in the mouse central nervous system. *Development* **125**, 2759–2770 (1998).
51. Abu-Khalil, A., Fu, L., Grove, E. A., Zecevic, N. & Geschwind, D. H. Wnt genes define distinct boundaries in the developing human brain: Implications for human forebrain patterning. *J. Comp. Neurol.* **474**, 276–288 (2004).
52. Grove, E. A., Tole, S., Limon, J., Yip, L. & Ragsdale, C. W. The hem of the embryonic cerebral cortex is defined by the expression of multiple Wnt genes and is compromised in Gli3-deficient mice. *Development* **125**, 2315–2325 (1998).
53. Theil, T., Alvarez-Bolado, G., Walter, A. & Ruther, U. Gli3 is required for Emx gene expression during dorsal telencephalon development. *Development* **126**, 3561–3571 (1999).
54. Bedogni, F. et al. Tbr1 regulates regional and laminar identity of postmitotic neurons in developing neocortex. *Proc. Natl Acad. Sci. USA* **107**, 13129–13134 (2010).
55. Hevner, R. F. et al. Tbr1 regulates differentiation of the preplate and layer 6. *Neuron* **29**, 353–366 (2001).
56. Pollen, A. A. et al. Molecular identity of human outer radial glia during cortical development. *Cell* **163**, 55–67 (2015).
57. Lewitus, E., Kelava, I. & Huttner, W. B. Conical expansion of the outer subventricular zone and the role of neocortical folding in evolution and development. *Front. Hum. Neurosci.* **7**, 424 (2013).
58. Torre-Ubieta, De. La., Won, L., Stein, H. & Geschwind, J. L. L. Advancing the understanding of autism disease mechanisms through genetics. *Nat. Med.* **22**, 345–361 (2016).
59. Takanaga, H. et al. Gli2 is a novel regulator of sox2 expression in telencephalic neuroepithelial cells. *Stem Cells* **27**, 165–174 (2009).
60. Friedrichs, M., Larralde, O., Skutella, T. & Theil, T. Lamination of the cerebral cortex is disturbed in Gli3 mutant mice. *Dev. Biol.* **318**, 203–214 (2008).
61. Iossifov, I. et al. The contribution of de novo coding mutations to autism spectrum disorder. *Nature* **515**, 216–221 (2014).
62. De Rubeis, S. et al. Synaptic, transcriptional and chromatin genes disrupted in autism. *Nature* **515**, 209–215 (2014).
63. Notwell, J. H. et al. TBR1 regulates autism risk genes in the developing neocortex. *Genome Res.* **26**, 1013–1022 (2016).
64. Cholfin, J. A. & Rubenstein, J. L. Frontal cortex subdivision patterning is coordinately regulated by Fgf8, Fgf17, and Emx2. *J. Comp. Neurol.* **509**, 144–155 (2008).
65. Semendeferi, K. et al. Spatial organization of neurons in the frontal pole sets humans apart from great apes. *Cereb. Cortex* **21**, 1485–1497 (2011).
66. McLean, C. Y. et al. GREAT improves functional interpretation of cis-regulatory regions. *Nat. Biotechnol.* **28**, 495–501 (2010).
67. Fromer, M. et al. De novo mutations in schizophrenia implicate synaptic networks. *Nature* **506**, 179–184 (2014).
68. Deciphering Developmental Disorders Study. Prevalence and architecture of de novo mutations in developmental disorders. *Nature* **542**, 433–438 (2017).
69. Kang, H. J. et al. Spatio-temporal transcriptome of the human brain. *Nature* **478**, 483–489 (2011).

## Acknowledgements

This work was supported by NIH grants 5R01MH060233; 5R01MH100027; 3U01MH103339; 1R01MH110927; 1R01MH094714 (to D.H.G.), R00MH113823 (to H. W.); NARSAD Young Investigator Award to H.W; Simons Foundation Powering Autism Research for Knowledge (SPARK) to H.W. We thank L. de la Torre-Ubieta, J.L. Stein, R. Bethlehem, M.J. Gandal, and Z. Nayernia for helpful discussions.

## Author contributions

H.W. carried out analysis and interpretation of the data. J.H. and C.K.O. performed functional validation experiment. C.L.H. helped designing the enrichment analysis. D.H. G. supervised the study. H.W. and D.H.G. wrote the manuscript.

## Additional information

**Competing interests:** The authors declare no competing interests.

