## [Peer Review File · Nature Communications]

Reviewers' comments:

Reviewer #1 (Remarks to the Author):

The authors have addressed the majority of my concerns. However, I have one remaining comment. In their HAR enrichment analyses, the authors are comparing HARs to a set of conserved elements, in order to account for any potential enrichment being due to the background conservation of HARs. This was in response to my previous comment. However, I am still not convinced this is the best approach. The authors choose a background set of conserved elements "larger than 20 bp with a phastCons score > 0.4."

First, it is unclear what the authors are doing here. phastCons provides an overall score for each conserved element it identifies, which can be expressed as a raw or transformed log-odds score. The authors do not appear to be using either of these. Instead, I think they are using phastCons scores for individual positions and only including conserved elements where all positions have a score > 0.4. I think the element-level scores are more appropriate.

Second, it is unclear if the constraint distributions of HARs and the background element set are the same. HARs are quite constrained overall, and the background set the authors have chosen may not reflect this. A more rigorous and conservative test would be to identify a superset of constrained elements with a similar constraint distribution as HARs, and that includes the HARs themselves. The authors could then conduct a permutation test where they randomly label constrained elements as HARs over some reasonable number of iterations (e.g, 10-20k), count the number of "HARs" associated with a particular feature or gene, and then calculate an empirical enrichment p value from the resulting distribution (based on the # of permutation iterations showing an equal or greater # of "HARs" associated with a feature compared to the observed #).

Reviewer #3 (Remarks to the Author):

The substantial revisions in response to both reviews have greatly improved the manuscript.

The new title is much better. However, the word "contribute" is a bit far-reaching since the study only shows causality in terms of gene expression changes in the CRISPR experiments with all other results being associations. It seems like the key points to make in the title are that the different sets of accelerated elements are in loci with distinct genes, and those loci are important in the CNS but have distinct associations with disease and cell types/states. Some further rewording would be appropriate.

Similarly, the word "orchestrate" in the abstract implies a causal role that is tempting to speculate about but not shown per se. Scanning the whole manuscript for language that suggests causality when the results show an association would be a good idea.

Supplemental Table 1 has the IDs of the genes mapped to each accelerated element, which is a useful resource. It would be helpful to also include a sheet from the gene perspective with common names for the genes or a column in this sheet that includes all those names separated by a delimiter. If there are particular loci responsible for any of the main enrichment results, naming

some of the genes may be interesting. For example, in Figure 3 a and b, what genes are in the tails of the distributions?

Regarding HAR names/IDs, it would be helpful to cross-reference the names used prior to the Doan et al. study, because functional studies have been published on about 90 HARs using those earlier names and readers will want to link the Doan results and your results to those prior studies. Names that include IDs from multiple papers separated by semi-colons can work, e.g, ANC152;BUSH08_5;HAR66 (as in Capra et al. PTRSB 2013 supplemental table 3 (supplemental file 5)). Another option is to use the Doan IDs but include a column in supplemental table 1 with the other IDs that map to each HAR you studied.

Typos:

- Abstract
- evolutionary  evolutionarily
- period at end of sentence for LoF and genes
- Fig 1d: hemopoiesis  hematopoiesis

It is our policy to sign reviews: Kathleen Keough & Katie Pollard

We appreciate the encouraging and constructive critiques from both reviewers. Addressing these has again substantially improved the manuscript, clarified its messages, and its connection to previous studies. Please find our point by point responses below:

Reviewer #1 (Remarks to the Author):

The authors have addressed the majority of my concerns. However, I have one remaining comment. In their HAR enrichment analyses, the authors are comparing HARs to a set of conserved elements, in order to account for any potential enrichment being due to the background conservation of HARs. This was in response to my previous comment. However, I am still not convinced this is the best approach. The authors choose a background set of conserved elements "larger than 20 bp with a phastCons score > 0.4."

First, it is unclear what the authors are doing here. phastCons provides an overall score for each conserved element it identifies, which can be expressed as a raw or transformed log-odds score. The authors do not appear to be using either of these. Instead, I think they are using phastCons scores for individual positions and only including conserved elements where all positions have a score > 0.4. I think the element-level scores are more appropriate.

Second, it is unclear if the constraint distributions of HARs and the background element set are the same. HARs are quite constrained overall, and the background set the authors have chosen may not reflect this. A more rigorous and conservative test would be to identify a superset of constrained elements with a similar constraint distribution as HARs, and that includes the HARs themselves. The authors could then conduct a permutation test where they randomly label constrained elements as HARs over some reasonable number of iterations (e.g, 10-20k), count the number of "HARs" associated with a particular feature or gene, and then calculate an empirical enrichment p value from the resulting distribution (based on the # of permutation iterations showing an equal or greater # of "HARs" associated with a feature compared to the observed #).

We thank the reviewer for this constructive comment. We used PhastCons scores because HARs were often defined by PhastCons scores (PMID: 29149249). We selected regions with a phstCons score > 0.4, because the majority of HARs also exhibited a phastCons score > 0.4 (Figure R1).

Figure R1. Distribution of phastCons scores for HARs (left) and phastCons- and size-matched regions (right).

However, we agree with the reviewer that it is reasonable to test enrichment with a more conservative default set of HARs. To make a comparable set, we randomly selected 10,000 sets of genomic regions that have size and phastCons score that are matched with HARs (hereby referred as “default regions”; phastCons score distribution of one set shown in Figure R1). We then overlapped these default regions to DNase I hypersensitive sites (DHS) in each tissue type and ran two types of enrichment analyses:

1. We checked the ratio of DHS overlap for 10,000 sets of default regions to generate a default background. We then compared this distribution with the ratio of DHS overlapping HARs. In both male and female fetal brain DHS datasets, the ratio of DHS overlapping HARs was much higher than the default distribution as shown below.

2. For each default dataset, we ran a Fisher’s exact test with the contingency table as follows:

# of DHS overlapping HARs	# of total HARs
# of DHS overlapping default regions	# of total default regions

We obtained an odds ratio (OR) for each Fisher’s exact test (which leads to a set of 10,000 ORs for each tissue type) and plotted these results. The result confirms that HARs show highest enrichment for DHS in fetal tissues, and more so in brain than in other tissue/cell types.

Collectively, these results show that the enrichment is robust to the background, even in the most conservative case. We now added this result in Supplementary Figure 1a.

Reviewer #3 (Remarks to the Author):

The substantial revisions in response to both reviews have greatly improved the manuscript.

The new title is much better. However, the word “contribute” is a bit far-reaching since the study only shows causality in terms of gene expression changes in the CRISPR experiments with all other results being associations. It seems like the key points to make in the title are that the different sets of accelerated elements are in loci with distinct genes, and those loci are important in the CNS but have distinct associations with disease and cell types/states. Some further rewording would be appropriate.

Similarly, the word “orchestrate” in the abstract implies a causal role that is tempting to speculate about but not shown per se. Scanning the whole manuscript for language that suggests causality when the results show an association would be a good idea.

We thank the reviewer for raising this point. We now changed the title to: “Different classes of human evolved regulatory elements show distinct patterns of association with brain evolution and disease.” We also changed the word “orchestrate” into “are associated with”.

Supplemental Table 1 has the IDs of the genes mapped to each accelerated element, which is a useful resource. It would be helpful to also include a sheet from the gene perspective with common names for the genes or a column in this sheet that includes all those names separated by a delimiter.

We thank the reviewer for this comment. We added a Supplementary Table (Supplementary Table 2) to describe all genes mapped to HARs, HGEs:FB, HGEs:AB, and HLEs.

If there are particular loci responsible for any of the main enrichment results, naming some of the genes may be interesting. For example, in Figure 3 a and b, what genes are in the tails of the distributions?

We selected a few genes for Figure 3b and added those genes: “HAR-associated genes with earlier breakpoints include *CPLX2*, whose protein product functions in synaptic vesicle exocytosis and *ITPR1*, which harbors mutations found in spinocerebellar ataxia... These genes include *S100B*, a marker for astrocytes.”

Regarding HAR names/IDs, it would be helpful to cross-reference the names used prior to the Doan et al. study, because functional studies have been published on about 90 HARs using those earlier names and readers will want to link the Doan results and your results to those prior studies. Names that include IDs from multiple papers separated by semi-colons can work, e.g., ANC152;BUSH08_5;HAR66 (as in Capra et al. PTRSB 2013 supplemental table 3 (supplemental file 5)). Another option is to use the Doan IDs but include a column in supplemental table 1 with the other IDs that map to each HAR you studied.

This is a great idea to facilitate comparisons with all of the literature – an oversight on our part. We now added the column to Supplementary Table 1.

In summary, we have responded to all of the issues raised by the reviewers and hope that the current manuscript is acceptable for publication.

Yours Sincerely,

Daniel H. Geschwind, MD, PhD
Gordon & Virginia MacDonald Distinguished Professor
Departments of Neurology, Psychiatry and Human Genetics

Reviewers' comments:

Reviewer #1 (Remarks to the Author):

My only remaining concern is that I am not sure the contingency table shown in point 2 of the rebuttal is correct. I think it should be:

HAR Background Row

DHS a b a+b

not DHS c d c+d

Column a+c b+d n

The authors should check this, otherwise I think the paper is suitable for publication.

Reviewer #3 (Remarks to the Author):

Regarding Reviewer 1's suggestion, we agree that the null distribution for enrichment analysis should be derived from the superset of phastCons elements from which HARs were derived so that they are matched to HARs. HARs were defined using phastCons elements identified via the Viterbi algorithm on alignments with the human sequenced masked. These were additionally filtered with constraints beyond GC content and conservation, such as excluding pseudogenes, filtering level 1 and 2 synteny, and more as listed at <http://docpollard.org/research/human-acceleration-in-mammal-conserved-elements/>. This method for defining phastCons elements and these constraints do not appear to have been considered in the enrichment analysis.

The title and claims in the paper are now more proportionate to the results shown.

Supplementary Table 2 would be improved by detailing which HAR is associated with which gene.

Common gene names would be more informative than Ensembl identifiers.

To ensure reproducibility, we recommend that supporting analysis code be made fully available and referenced in the text. All data used and generated, including data generated during analysis steps, should be released or should be easy to generate from provided data plus freely available code. There is a small example piece of code provided ("187939_1_data_set_3485546_pkrttm.txt"), but it is not comprehensive of the entire analysis.

It is our policy to sign reviews: Kathleen Keough & Katie Pollard

We thank the reviewers for their very helpful guidance and critiques, as well as their positive sentiments about the work. We have addressed all of the reviewers concerns and hope that the manuscript is now acceptable for publication in *Nature Communications*. Please find our response to the reviewers below.

Reviewer #1 (Remarks to the Author):

My only remaining concern is that I am not sure the contingency table shown in point 2 of the rebuttal is correct. I think it should be:

HAR Background Row
 DHS a b a+b
 not DHS c d c+d
 Column a+c b+d n

The authors should check this, otherwise I think the paper is suitable for publication.
 → We thank the reviewer for catching this. We now updated our contingency table as follows:

	DHS	Not DHS	Row Total
HAR	# of HARs that overlap with DHS in a given cell type = A	# of HARs that do not overlap with DHS in a given cell type = D	A+C
Background (evolutionary conserved regions)	# of evolutionary conserved regions that overlap with DHS in a given cell type.= B	# of evolutionary conserved regions that do not overlap with DHS in a given cell type = D	B+D
Column Total	A+B	C+D	N (=A+B+C+D)

Reviewer #3 (Remarks to the Author):

Regarding Reviewer 1's suggestion, we agree that the null distribution for enrichment analysis should be derived from the superset of phastCons elements from which HARs were derived, so that they are matched to the appropriate HARs. HARs were defined using phastCons elements identified via the Viterbi algorithm on alignments with the human sequenced masked. These were additionally filtered with constraints beyond GC content and conservation, such as excluding pseudogenes, filtering level 1 and 2 synteny, and more as listed at <http://docpollard.org/research/human-acceleration-in-mammal-conserved-elements/>. This method for defining phastCons elements and these constraints do not appear to have been considered in the enrichment analysis.

→ We agree with the reviewer that building the null distribution is an important issue. However, we compiled HARs that were defined by multiple groups, where each set was

defined by different statistical tests, filters, and multiple species alignments (as mentioned in Capra et al., 2015). Therefore, using only one way of defining the background may not affect equally HARs defined by other groups. Instead, we used three different ways to define the null distribution: (1) GC content matched genomic regions, (2) genomic regions with evolutionary conservation (phastCons score >0.4), and (3) phastCons score matched regions with HARs. Regardless of the null distribution, we observed robust enrichment for fetal brain DHS, supporting the robustness of our results. Moreover, this result is in line with Capra et al., 2015, where they identified enrichment of HARs in fetal active regulatory elements.

Thanks to the reviewer's point, we now mentioned in the manuscript that we used phastCons score calculated by <http://compugen.cshl.edu/phast/>, using the R library code: *phastCons100way.UCSC.hg19*.

The title and claims in the paper are now more proportionate to the results shown.
→ We thank the reviewer for this comment.

Supplementary Table 2 would be improved by detailing which HAR is associated with which gene.
→ We already have the information in Supplementary Table 1. Supplementary Table 1 has HARs that are mapped to each gene in Hi-C data from fetal brain and adult brain.

Common gene names would be more informative than Ensembl identifiers.
→ Thanks. We also added Human gene symbols along with Ensembl gene IDs.

To ensure reproducibility, we recommend that supporting analysis code be made fully available and referenced in the text. All data used and generated, including data generated during analysis steps, should be released or should be easy to generate from provided data plus freely available code. There is a small example piece of code provided ("187939_1_data_set_3485546_pkrttm.txt"), but it is not comprehensive of the entire analysis.
→ We appreciate this comment and apologize for not including all of the code. We now submitted all of the code used for the analyses (codes.R).

REVIEWERS' COMMENTS:

Reviewer #3 (Remarks to the Author):

All of our comments have been addressed.

It is our policy to sign reviews: Kathleen Keough & Katie Pollard